# VER🐦: Unifying Verbalizing Entities and Relations

**Jie Huang**    **Kevin Chen-Chuan Chang**

Department of Computer Science, University of Illinois at Urbana-Champaign

{jeffhj, kcchang}@illinois.edu

## Abstract

Entities and relationships between entities are vital in the real world. Essentially, we understand the world by understanding entities and relations. For instance, to understand a field, e.g., *computer science*, we need to understand the relevant concepts, e.g., *machine learning*, and the relationships between concepts, e.g., *machine learning* and *artificial intelligence*. To understand a person, we should first know who he/she is and how he/she is related to others. To understand entities and relations, humans may refer to natural language descriptions. For instance, when learning a new scientific term, people usually start by reading its definition in dictionaries or encyclopedias. To know the relationship between two entities, humans tend to create a sentence to connect them. In this paper, we propose **VER🐦**: a unified model for **V**erbalizing **E**ntities and **R**elations. Specifically, we attempt to build a system that takes any entity or entity set as input and generates a sentence to represent entities and relations. Extensive experiments demonstrate that our model can generate high-quality sentences describing entities and entity relationships and facilitate various tasks on entities and relations, including definition modeling, relation modeling, and generative commonsense reasoning.[1]

## 1 Introduction

*What is X? What is the relationship between X and Y?* We come up with these questions almost every day. When we come across a new term, e.g., *twin prime*, we usually refer to its definition to understand it, i.e., "*A **twin prime** is a prime number that is either 2 less or 2 more than another prime number*". To express the understanding about relationship between entities (e.g., *carbon dioxide* and *water*), we create a sentence to represent their relationship: "***Carbon dioxide** is soluble in **water***".

Basically, we understand entities and relations by "*verbalizing*" them. Verbalizing entities and relations also tests our knowledge about entities and relations. Literally, by verbalizing entities and relations, we understand the world.

Similarly, *do machines have the ability to verbalize entities and relations? Can machines learn about entities and relations from verbalizing them?* The answer is "*Yes*". Recent studies show that by giving the surface name of an entity (and its context), models (after training) can generate coherent sentences to represent it, i.e., *definition modeling* (Noraset et al., 2017; Gadetsky et al., 2018; Bevilacqua et al., 2020; August et al., 2022; Huang et al., 2022b; Gardner et al., 2022), and by giving the surface names of a pair of entities, machines can generate coherent sentences describing their relationships, i.e., *(open) relation modeling* (Huang et al., 2022a,c). However, verbalizing entities requires understanding relationships between entities, and verbalizing entity relationships requires understanding entities themselves, while existing works deal with entity and relation verbalization separately, ignoring the connections between them.

Besides, recent works (Devlin et al., 2019; Lewis et al., 2020; Radford et al., 2019; Brown et al., 2020) have shown that language models pre-trained with self-supervised objectives can equip the model with a significant amount of knowledge (Petroni et al., 2019; Roberts et al., 2020) and achieve substantial gains after fine-tuning on a specific task. Can we continually pre-train the models with pre-training objectives on entities and relations to enhance their ability on verbalizing entities and relations? In this way, the model can be easier and better adapted to specific tasks on entities and relations and even be used without additional training.

Therefore, we aim to solve entity and relation verbalization in a unified form and pre-train a model for entity and relation understanding. Essentially, definition modeling and relation model-

---

[1] Code, data, and the trained models are available at https://github.com/jeffhj/VER.

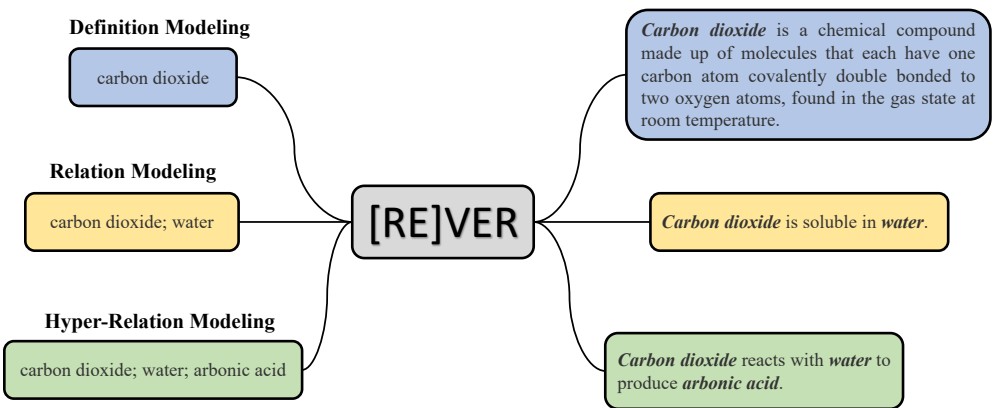

Figure 1: A diagram of [RE]VER. We feed the model with entity(s) and train it to reconstruct sentences containing all the entities. This allows us to use a single model to better "*verbalize*" entities and complex entity relationships.

ing can be unified as an "entity(s) → sentence" task, i.e., given a set of entities, generating a sentence describing the entities and their relationships. When the size of the set is 1, it is equivalent to definition modeling, and when the size of the set is 2, it is equivalent to relation modeling. By defining the task in this form, we can even model more complex relationships among entities since entity relationships can go beyond pairwise (Bretto, 2013), named *hyper-relation modeling*, e.g., {*carbon dioxide*, *water*, *arbonic acid*} → "***Carbon dioxide** reacts with **water** to produce **arbonic acid**". Based on this, we propose **VER**🔊 (pronunciation: /vɜː/): a unified model for **V**erbalizing **E**ntities and **R**elations (Figure 1). Specifically, we pre-train models by forming a self-supervised text reconstruction task: given an entity or a set of entities, reconstruct the original sentences (e.g., a definition or a relation description) containing them in the training corpus. In this way, the models acquire knowledge about entities and relations and learn to connect entities to a meaningful coherent sentence. Since implicit knowledge stored in the parameters of the models may not be sufficient for generating meaningful sentences for the target entities, we also study **VER** in the retrieval-enhanced setting (Guu et al., 2020; Izacard et al., 2022; Huang et al., 2023b), named **REVER**, i.e., **R**etrival-**E**nhanced **VER**, by pre-training models augmented with sentences contained the target entities in the pre-training corpus. Throughout the remainder of this paper, we use "**[RE]VER**" to denote both VER and REVER.

Experiments on six datasets demonstrate the superiority of our model in verbalizing entities and relations. Especially in low-resource settings, [RE]VER can achieve significantly better results than baselines on definition modeling, relation modeling, and generative commonsense reasoning. In addition, the performance of [RE]VER without additional training is impressive, making itself a potential knowledge source of entities and relations, which may benefit tasks on entities and relations such as entity typing (Ren et al., 2016), relation extraction (Bach and Badaskar, 2007), and knowledge graph completion (Lin et al., 2015).

## 2 Background and Formulations

**Definition Modeling**. Definition modeling aims to generate definitions of entities, which can be formulated as a conditioned sequence generation task. For instance, given *twin prime*, the expected output is the definition of *twin prime*: "A *twin prime* is a prime number that is either 2 less or 2 more than another prime number". We follow the standard sequence-to-sequence formulation in Noraset et al. (2017); Huang et al. (2022b): given entity $x$, the probability of the generated definition $s = [s_1, \ldots, s_m]$ is computed auto-regressively:

$$P(s|x) = \prod_{i=1}^{m} P(s_i|s_0, s_1, \ldots, s_{i-1}, x), \quad (1)$$

where $m$ is the length of $s$, $s_i$ is the $i$th token of $s$, and $s_0$ is a special start token.

**(Open) Relation Modeling**. Relation modeling attempts to generate coherent and meaningful sentences describing relationships between entities, where types of relations are not pre-specified, i.e., in an "*open*" setting (Huang et al., 2022a). For example, for *carbon dioxide* and *water*, their relationship can be described as "*Carbon dioxide is soluble in water.*" For *machine learning* and *algorithm*, the expected output could be "*Machine*

*learning* explores the study and construction of *algorithms* that can learn from and make predictions on data." Formally, given entity pair $(x, y)$, the probability of the generated relation description $s = [s_1, \ldots, s_m]$ is calculated as:

$$P(s|x,y) = \prod_{i=1}^{m} P(s_i|s_0, s_1, \ldots, s_{i-1}, x, y). \quad (2)$$

**Hyper-Relation Modeling (Unified Form).** Previous works mainly focus on verbalizing single entities or entity pairs. However, in the real world, relationships between entities can be more complex – beyond pairwise, named "hyper" relationships (Bretto, 2013; Tu et al., 2018; Huang et al., 2019, 2020). For example, "*carbon dioxide* reacts with *water* to produce *carbonic acid*". Here, there are tuplewise relationships among *carbon dioxide*, *water*, and *carbonic acid*. Verbalization of hyper relationships was initially investigated in Common-Gen (Lin et al., 2020) but was limited to commonsense concepts, and their outputs are simple short sentences describing everyday scenarios containing the given concepts. We attempt to model and verbalize more general complex "hyper" relationships among entities and find a unified framework to combine single entities (1 entity), pairwise relationships (2 entities), and "hyper" relationships ($\geq$3 entities). Combining with definition modeling and relation modeling, we adopt the following unified form:

$$P(s|\mathcal{E}) = \prod_{i=1}^{m} P(s_i|s_0, s_1, \ldots, s_{i-1}, \mathcal{E}), \quad (3)$$

where $\mathcal{E}$ is the entity set and $|\mathcal{E}| \geq 1$.

# 3  [RE]VER🔊: Verbalizing Entities and Relations

To verbalize an entity, we are likely to connect it to other entities, which requires knowledge about entity relationships. To understand entity relationships, we need to know about entities first. Based on this, we attempt to verbalize entities and relations in a unified form and propose **[RE]VER🔊**: a unified model for **V**erbalizing **E**ntities and **R**elations. We first create a large dataset with the formulation in Eq. (3), and pre-train a model on this dataset, which equips the model with a significant amount of knowledge about entities and relations and enables the model to generate coherent and meaningful sentences connecting the entities. The model can be further fine-tuned on specific datasets, e.g., definition modeling,

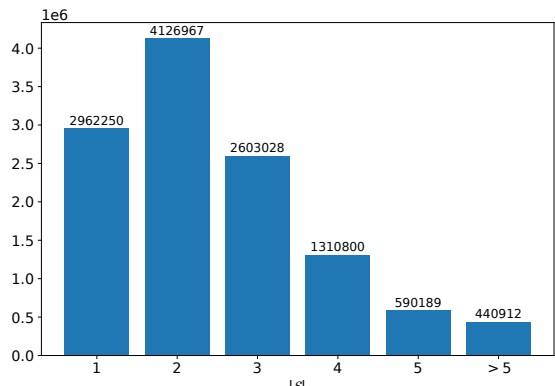

Figure 2: Statistics of the pre-training data.

relation modeling, and generative commonsense reasoning, to achieve better performance on specific tasks.

## 3.1  WiV Data

We prepare the pre-training data with Wikipedia. Wikipedia is a large encyclopedia containing a huge number of entities. Wikipedia is well maintained and the content is generally of high quality. We extract entity sets and sentences from Wikipedia. Specifically, we use the 2022-08-01 dump[2] of English Wikipedia. For each page, we extract the plain text by WikiExtractor[3]. To find expressions that refer to the same entity, we use the neuralcoref (Clark and Manning, 2016) coreference resolution tool in spaCy[4] to preprocess the documents. Since we would like the model to capture the main characteristics of entities and relations, we take the first 5 sentences from each page (those sentences are usually definitional sentences or sentences expressing entity relationships). To identify entities, we utilize the hyperlinks in each Wikipedia page to build a local mention-entity mapping. And then, we process each sentence and extract the corresponding entity set based on the mention-entity mapping. In this way, we build mapping $\mathcal{E} \rightarrow s$, e.g., "{*Data mining, data sets, machine learning, statistics, database systems*} $\rightarrow$ *Data mining is the process of extracting and discovering patterns in large data sets involving methods at the intersection of machine learning, statistics, and database systems*." Since for a single entity, we prefer the model to generate a definition-like sentence rather than a random sentence including it, we collect the first sentence on each page and collect the input-output pair as "{*[page title]*} $\rightarrow$ *1st sentence*", e.g.,

[2] https://dumps.wikimedia.org/enwiki/20220801/
[3] https://github.com/attardi/wikiextractor
[4] https://spacy.io/

"{*deep learning*} → *Deep learning is part of a broader family of machine learning methods based on artificial neural networks with representation learning*." We filter out input-output pairs where $|\mathcal{E}| = 1$ and $s \neq$ *1st sentence*. We keep out pages appearing in the validation and test sets of Huang et al. (2022b,a) and filter out entity sets appearing in the datasets of Huang et al. (2022b); August et al. (2022); Huang et al. (2022a); August et al. (2022); Lin et al. (2020). We call this dataset **WiV** (**Wi**kipedia **VER**). The number of training examples with different sizes of entity sets is summarized in Figure 2.

## 3.2 Model

At a high level, we pre-train a model by training it to reconstruct target sentences conditioned on the entity set with the pre-training data. Specifically, for VER, we continually pre-train BART (Lewis et al., 2020) on the data constructed in Section 3.1. BART adopts a transformer-based encoder-decoder architecture with input text fed to the encoder and output text produced by the decoder. For our continual pre-training, we encode entity set $\mathcal{E} = \{e_1, e_2, \ldots, e_{|\mathcal{E}|}\}$ to sequence "$e_1$; $e_2$; $\ldots$; $e_{|\mathcal{E}|}$", e.g., {*carbon dioxide*, *water*, *carbonic acid*} to "carbon dioxide; water; carbonic acid". Here we keep the order of entities as the order they appear in the sentence. We choose this design because different orders may correspond to different natural language descriptions (e.g., the descriptions are different when an entity is used as subject vs. object). We would like to mention that although we keep the order here, the model can deal with inputs with random entity orders after fine-tuning (e.g., CommonGen (Lin et al., 2020) as shown in Section 4.4). We train two versions of the model: *VER-base* with 6 layers in the encoder and decoder, and *VER-large* with 12 layers in each, corresponding to BART-based and BART-large respectively.

For REVER, given an input with entity set $\mathcal{E} = \{e_1, e_2, \ldots, e_{|\mathcal{E}|}\}$, we sample sentences containing the target entities from WiV. Specifically, we search the dataset to find sentences whose entities overlap most with the given entity set repeatedly (the target sentence is excluded). In order for the model to be able to handle retrieved knowledge of different lengths, for each input, we set the maximum number of retrieved sentences as a random number $h$ from 0 to 10. With the retrieved sentences $s'_1, s'_2, \cdots, s'_h$, we encode the input as "$e_1$; $e_2$;

$\ldots$; $e_{|\mathcal{E}|}$ [SEP] $s'_1$ [SEP] $s'_2$ [SEP] $\ldots$ [SEP] $s'_h$". Similar to VER, we continually pre-train BART-base and BART-large on the input-output pairs and get two versions of the model: *REVER-base* and *REVER-large*.

## 3.3 Training Process

We pre-train [RE]VER-large and [RE]VER-base with the fairseq library[5]. We use Adam with $\beta_1 = 0.9$, $\beta_2 = 0.999$, and $\epsilon = 10^{-8}$, and set the clip threshold of gradients as $0.1$. All models use weight decay of $0.001$ and dropout of $0.1$. We set the learning rate as $5 \times 10^{-5}$ and use batch size of $1,024$ tokens, updating every 16 iterations. We set the number of warmup steps as $1,000$. We use a small validation set to examine whether the training converges. All models were trained on NVIDIA A100 GPUs or A40 GPUs, and the training converged in 60 epochs.

# 4 Experiments

In this section, we evaluate [RE]VER on definition modeling, relation modeling, and hyper-relation modeling in three settings: 1) fine-tune the model on the full task-specific training data; 2) fine-tune the model in low-resource settings; 3) directly use the model without fine-tuning. The main goal of the experiments is to verify whether the continual pre-training step can enhance models' ability on verbalizing entities and relations with/without external knowledge.

## 4.1 Experimental Setup

**Datasets**. For definition modeling, we use the datasets of **UJ-CS/Math/Phy** (Huang et al., 2022b) and **Sci & Med** (August et al., 2022). For relation modeling, we use the dataset built in Huang et al. (2022a) (**ORM**), we take the filtered test set for evaluation since the quality is higher. For hyper-relation modeling, there is no existing dataset. We find that **CommonGen** (Generative Commonsense Reasoning) (Lin et al., 2020) can serve our purpose for evaluation since the task formulation is similar: given a set of common concepts, i.e., $\mathcal{E}$ ($3 \leq |\mathcal{E}| \leq 5$), generating a coherent sentence describing an everyday scenario using these concepts. By testing on CommonGen, we can also measure the ability of our model for domain adaptation. Since the reference sentences of the official test set of CommonGen are not released, for the full

---

[5] https://github.com/facebookresearch/fairseq

|       | UJ-CS | | | | UJ-Math | | | | UJ-Phy | | | | Sci & Med | | | |
| *100%* | **BL** | **R-L** | **MT** | **BS** | **BL** | **R-L** | **MT** | **BS** | **BL** | **R-L** | **MT** | **BS** | **BL** | **R-L** | **MT** | **BS** |
|---|---|---|---|---|---|---|---|---|---|---|---|---|---|---|---|---|
| BART  | 8.31 | 28.02 | 12.83 | 77.97 | 6.89 | 28.50 | 10.97 | 76.45 | 5.28 | 25.75 | 10.57 | 76.88 | 13.13 | 31.75 | 13.30 | 79.31 |
| VER   | 8.43 | 30.11 | 13.06 | 79.57 | 7.09 | 31.94 | 11.86 | 78.07 | 7.09 | 30.63 | 12.71 | 79.18 | 13.95 | 33.57 | 14.84 | 80.49 |
| SOTA  | 22.66 | 38.12 | 20.30 | 82.00 | 23.22 | 39.39 | 19.61 | 80.30 | 20.84 | 37.66 | 19.26 | 81.18 | 20.55 | 37.70 | 19.24 | 81.98 |
| REVER | **23.04** | **38.85** | **20.52** | **82.79** | **23.25** | **41.95** | **20.49** | **81.61** | **21.92** | **38.01** | **19.76** | **81.94** | **21.29** | **38.14** | **19.95** | **82.55** |
| *10%* | **BL** | **R-L** | **MT** | **BS** | **BL** | **R-L** | **MT** | **BS** | **BL** | **R-L** | **MT** | **BS** | **BL** | **R-L** | **MT** | **BS** |
| BART  | 3.50 | 22.98 | 8.68 | 75.55 | 4.32 | 25.42 | 8.94 | 75.21 | 3.27 | 24.19 | 8.43 | 75.72 | 5.56 | 23.97 | 9.47 | 77.13 |
| VER   | 6.43 | 28.24 | 12.36 | 78.77 | 7.24 | 31.18 | 11.79 | 77.82 | 6.43 | 30.57 | 12.42 | 78.92 | 7.59 | 28.25 | 12.09 | 78.70 |
| SOTA  | 17.48 | 32.32 | 17.39 | 80.60 | 19.74 | 36.81 | 18.02 | 79.85 | 16.82 | 32.83 | 16.96 | 79.86 | 12.99 | 30.82 | 14.88 | 80.42 |
| REVER | **17.71** | **34.38** | **18.14** | **81.63** | **21.00** | **38.83** | **19.07** | **80.85** | **20.32** | **36.82** | **19.16** | **81.59** | **15.46** | **33.34** | **16.83** | **80.86** |
| *0%* | **BL** | **R-L** | **MT** | **BS** | **BL** | **R-L** | **MT** | **BS** | **BL** | **R-L** | **MT** | **BS** | **BL** | **R-L** | **MT** | **BS** |
| VER$^-$ | 4.81 | 26.24 | 11.62 | 77.55 | 6.00 | 30.57 | 11.41 | 77.35 | 5.70 | 28.62 | 12.12 | 78.06 | 5.98 | 22.84 | 11.01 | 75.32 |
| VER   | 5.05 | 26.55 | 11.96 | 77.84 | 6.33 | 30.36 | 11.57 | 76.88 | 5.95 | 28.79 | 12.35 | 78.13 | 6.06 | **23.49** | 11.22 | 75.49 |
| REVER | **10.38** | **30.36** | **14.85** | **79.98** | **11.29** | **35.09** | **14.60** | **79.66** | **12.68** | **34.49** | **16.10** | **80.83** | **12.13** | 23.47 | **13.63** | **76.53** |

Table 1: Results of definition modeling.

data setting, we submit the results generated by the model to the leaderboard to get the performance. For the low-resource settings, we use the in-house split presented in Wang et al. (2022) to facilitate comparison between our model and the baseline.

**Baselines**. Since [RE]VER is trained based on BART (Lewis et al., 2020), we include BART as a baseline for all the tasks. We also compare with SOTA of each task. For definition modeling, the SOTA is CDM-S5,C5 proposed by Huang et al. (2022b), which leverages definitional sentences retrieved by two definition extractors in its generation. For relation modeling, we compare with the best model in Huang et al. (2022a), which incorporates reasoning paths in knowledge graphs to generate entity relation descriptions. For Common-Gen, we compare with DKMR[2] (He et al., 2022) (SOTA), RACo (Yu et al., 2022) (runner-up), and RE-T5 (Wang et al., 2021) in the leaderboard.

**Metrics**. For definition modeling and relation modeling, we follow Huang et al. (2022b,a) to use BLEU (BL) (Papineni et al., 2002)[6], ROUGE-L (R-L) (Lin, 2004), METEOR (MT) (Banerjee and Lavie, 2005), and BERTScore (BS) (Zhang et al., 2019) for automatic evaluation. Among them, BLEU, ROUGE-L, and METEOR focus on measuring surface similarities by n-gram overlap, and BERTScore is based on the similarities of contextual token embeddings. For the evaluation on generative commonsense reasoning, we follow Lin et al. (2020) to use BLEU-4, CIDEr (Vedantam et al., 2015), and SPICE (Anderson et al., 2016), where

CIDEr and SPICE focus on evaluating the concept association instead of n-gram overlap. We also sample 100 examples for test sets and ask three human annotators (graduate students in computer science) to evaluate the generated outputs with a 1-5 rating scale used in Ishiwatari et al. (2019) (for definition modeling) and Huang et al. (2022c) (for relation modeling).

**Implementation details**. For each task, to make the results comparable and reproducible, we adopt the same hyperparameters as the implementation of the authors to fine-tune BART. We also use the same hyperparameters as BART to fine-tune [RE]VER on specific tasks. For definition modeling, since Huang et al. (2022b) use BART-base, we fine-tune [RE]VER-base for a fair comparison. For relation modeling and generative commonsense reasoning, we use [RE]VER-large. To acquire sentences as the external knowledge for REVER, we use the same knowledge leveraged by CDM-S5,C5 (Huang et al., 2022b) for definition modeling. For relation modeling, we retrieve sentences from WiV as described in Section 3.2. For CommonGen, we leverage the knowledge retrieved by Matching Retriever of RE-T5 (Wang et al., 2021). For the low-resource settings, we randomly sample the corresponding number of training samples from the train sets. For all the models and settings, we train the models with enough epochs to ensure the training converges and select the checkpoint with the best validation performance.

### 4.2 Definition Modeling

In Table 1, we report the results of definition modeling on the four datasets. For the full-data set-

---

[6]The version implemented on https://github.com/mjpost/sacrebleu.

ting (*100%*), VER outperforms BART on all the datasets. In the low-resource setting (*10%*, i.e., fine-tune the model with only 10% of the data), we observe that VER achieves a more significant improvement. Besides, by leveraging the same external knowledge of the SOTA (Huang et al., 2022b), REVER can outperform the SOTA. The results demonstrate that after continually pre-training the model with the entity(s)-to-sentence reconstruction task, the model acquires more knowledge about entities and has a better ability to verbalize entities.

Since [RE]VER can generate a sentence (possibly a definitional sentence) by taking any entity as input without fine-tuning, we also report the "0%" results, where no training data on the specific tasks are

| | Score (1-5) |
|---|---|
| BART | 1.13 |
| VER | 2.25 |
| SOTA | 3.98 |
| REVER | **4.51** |

Table 2: Averaged human annotated scores on UJ-CS (10% training data).

used to fine-tune the model. We find that VER (*0%*) can achieve better performance than BART (*10%*) and REVER can even outperform BART trained with full data, which indicates the strong performance of [RE]VER on definition modeling without additional training.

To validate whether the joint training of relation modeling will benefit or harm the performance of definition modeling, we train a version of VER only with data examples where $|\mathcal{E}| = 1$ (VER$^-$). From the results in Table 1, we observe that the performance of VER$^-$ is slightly worse than VER, which means relation understanding by relation modeling and hyper-relation modeling can benefit (at least does not harm) definition modeling.

Table 2 presents the human evaluation results. We observe that when trained with only 10% of the training data (1173 examples), BART struggles to generate meaningful definitions, with most of its attempts receiving a score of 1. VER, while able to produce some meaningful definitions, still falls short of the desired quality, as many of the definitions it generates contain errors. On the other hand, REVER performs remarkably well, achieving an average score of 4.51. This is a significant leap over SOTA results. It demonstrates that, even in the low-resource setting, REVER can generate definitions of exceptional quality. This underscores the importance of both pre-training and retrieved knowledge for generating definitions of entities.

## 4.3 (Open) Relation Modeling

Figure 3 summarizes the results of open relation modeling. We observe that [RE]VER consistently outperforms SOTA on the four metrics, and the performance improvement is more significant when the model is fine-tuned on less training data. We also find that the performance of the model without any additional fine-tuning (*# Training Examples =* 0) is quite impressive. On two metrics (R-L and MT), the performance is even better than BART trained with 50,000 examples.

The results indicate that rich entity and relational knowledge are learned by [RE]VER through continual pre-training. Besides, the text reconstruction task enables the model to produce natural language decriptions of relations by connecting entities in a coherent sentence.

Table 3 showcases the human evaluation results for open relation modeling. Remarkably, both VER and REVER significantly outperform the state-of-the-art in a low-resource setting

| | Score (1-5) |
|---|---|
| BART | 2.05 |
| VER | 2.79 |
| SOTA | 2.23 |
| REVER | **3.67** |

Table 3: Averaged human scores on ORM (500 training examples).

(trained with only 500 examples). However, we do note that [RE]VER still grapples with hallucination, leading to inaccuracies in the generated relation descriptions. For instance, some outputs wrongly state a location to be in a different city, albeit in the correct state and country. Nonetheless, considering the model is trained with just 500 examples, the results are still quite impressive.

## 4.4 Hyper-Relation Modeling (Generative Commonsense Reasoning)

Table 4 reports the CommonGen leaderboard results of [RE]VER and baselines. We find that although the style of sentences used to pre-train [RE]VER is quite different from that in Common-Gen, e.g., "A *dog* leaps to *catch* a *thrown frisbee*", the continual pre-training step still benefits the generative commonsense reasoning ability of the model. Besides, we observe that REVER outperforms RACo on two metrics, despite the external knowledge base used in REVER (same as RE-T5) is much smaller than RACo.

From the results of low-resource experiments in Figure 4, we observe that the improvement of

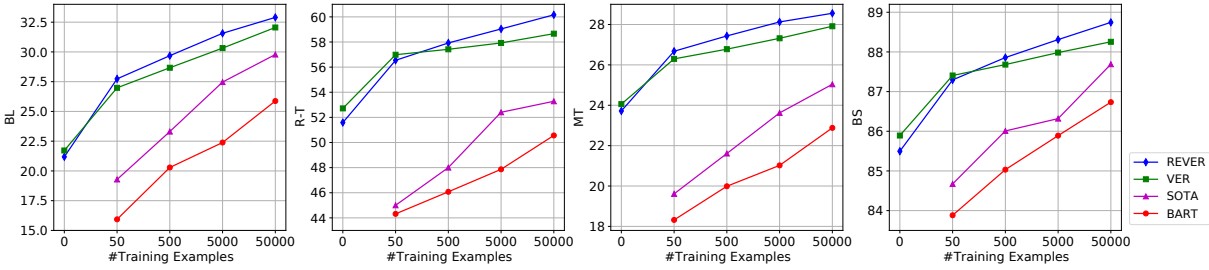

Figure 3: Results of open relation modeling (ORM) with different numbers of training examples.

| | BLEU-4 | CIDEr | SPICE |
|---|---|---|---|
| BART | 31.83 | 13.98 | 28.00 |
| VER | 34.22 | 16.28 | 28.28 |
| RE-T5 | 40.86 | 17.66 | 31.08 |
| RACo (runner-up) | 43.12 | 19.14 | 34.03 |
| DKMR[2] (SOTA) | **44.33** | **19.54** | **34.59** |
| REVER | 43.55 | 19.19 | 33.70 |

Table 4: Results on CommonGen (leaderboard v1.1). The best results are **bold** and second best ones are underlined.

[RE]VER in the low-resource settings is very significant, despite the style of sentences in the pre-training and fine-tuning being quite different. Although the zero-shot performance of [RE]VER on CommonGen is poor, with 50 training examples, [RE]VER can achieve better results than BART trained with 5,000 training examples on CIDEr and SPICE (according to Lin et al. (2020), SPICE and CIDEr correlate the human evaluation the most). This indicates the domain adaptation ability of [RE]VER is high.

In Table 5, we present some sample outputs of the models. Here $\{dog, frisbee, catch, throw\}$ is an example in the test set of CommonGen that is used as the demonstration in their paper. From the results, we find that despite all the baselines failing in this example, VER and REVER both produce a plausible sentence describing a correct everyday scenario using all the concepts. We also find that the performance of VER in low-resource settings is very impressive. For instance, BART trained with 5,000 training examples cannot even generate a sentence containing all the concepts, and the generated sentence describes a weird scenario, while VER trained with only 50 examples can generate a coherent sentence containing all the concepts. Besides, without fine-tuning, BART cannot generate anything meaningful, while [RE]VER can still generate a reasonable sentence using all the concepts, although the style of the sentence is different from the ground truth.

## 5 Related Work

**Definition Modeling**. *Definition modeling* aims to generate a definition for a given entity/term. This problem was first studied in Noraset et al. (2017) in a form of generating definitions of words with word embeddings. Later works focus on generating definitions with contexts or external knowledge (Gadetsky et al., 2018; Ishiwatari et al., 2019; Washio et al., 2019; Mickus et al., 2019; Li et al., 2020; Reid et al., 2020; Bevilacqua et al., 2020; Huang et al., 2021, 2022b; August et al., 2022). For instance, Bevilacqua et al. (2020) fine-tune BART (Lewis et al., 2020) on the word/phrase-definition pairs with given contexts. August et al. (2022) aim to control the complexity of the definition while generating the definition for a given term. Huang et al. (2022b) propose to combine definition extraction and definition generation to improve the performance of definition modeling.

**(Open) Relation Modeling**. *Open Relation Modeling* (Huang et al., 2022a) aims to generate a sentence to describe the relationship within a given entity pair. The authors propose to fine-tune BART and incorporate reasoning paths in knowledge graphs as auxiliary knowledge to solve this task. As follow-up work, Huang et al. (2022c); Zhu et al. (2023) construct *Descriptive Knowledge Graph* by extracting and generating sentences explaining entity relationships with the analysis of dependency patterns and a transformer-based relation description synthesizing model.

**Generative Commonsense Reasoning.** *Generative Commonsense Reasoning* (Lin et al., 2020; Liu et al., 2023) is a constrained text generation task that tests machines' ability to generate a coherent sentence describing everyday scenarios containing the given concepts. Later works mainly focus on improving performance by retrieving external knowledge to help the generation. For instance, KG-BART (Liu et al., 2021) designs a knowledge

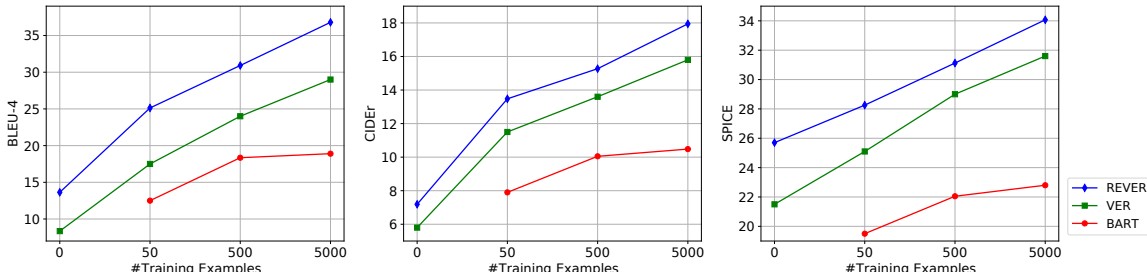

Figure 4: Results of the low-resource experiments on CommonGen (in-house) with different numbers of training examples. Since the code of the SOTA, i.e., DKMR$^2$, is not released, we do not report its performance here. For comparison in the full data setting, please refer to Table 4.

| Concepts | {*dog, frisbee, catch, throw*} |
|---|---|
| Human 1 | A *dog* leaps to *catch* a *thrown frisbee*. |
| Human 2 | The *dog catches* the *frisbee* when the boy *throws* it. |
| Human 3 | A man *throws* away his *dog* 's favorite *frisbee* expecting him to *catch* it in the air. |
| GPT-2 | A *dog throws* a *frisbee* at a football player. |
| UniLM | Two *dogs* are *throwing frisbees* at each other . |
| BART | A *dog throws* a *frisbee* and a *dog catches* it. |
| T5 | *dog catches* a *frisbee* and *throws* it to a *dog* |
| VER | A man is *throwing* a *frisbee* to his *dog*, who *catches* it. |
| BART | (0) | ;; |
| VER | (0) | a *dog* that is trained to *throw* and retrieve a *frisbee* by its handler is given the task of making a *catch* and *throw* of the disc. |
| BART | (50) | A boy is playing *frisbee* with his friends |
| VER | (50) | a *dog catches* a *frisbee* and *throws* it to a person. |
| BART | (500) | A *dog catches* a *frisbee* during a football game. |
| VER | (500) | A *dog catches* a *frisbee* and *throws* it. |
| BART | (5000) | A man is *throwing* a *frisbee* to a woman who is *catching* it. |
| VER | (5000) | Two *dogs* are playing *frisbee* and one of them is *catching* and *throwing* it. |
| REVER | (0) | The man begins to *throw* the *frisbee*, and the *dog* jumps into the air to *catch* it. |
| REVER | (0) | A man *throwing* a *frisbee* and his *dog catching* it. |

Table 5: Sentences produced by commonly-used pre-trained models and [RE]VER. VER (50) refers to VER fine-tuned with 50 training examples. Here we take the example in the demonstration of Lin et al. (2020).

graph-augmented model that incorporates the embeddings of relations of concepts from ConceptNet (Speer et al., 2017) as auxiliary inputs of BART. EKI-BART (Fan et al., 2020), Re-T5 (Wang et al., 2021), KFCNet (Li et al., 2021), DKMR$^2$ (He et al., 2022), and RACo (Yu et al., 2022) retrieve prototype sentences from external corpora as auxiliary input to language models such as BART and T5 (Raffel et al., 2020). In this work, we show that continual training on verbalizing entities and relations can improve models' generative commonsense reasoning ability, either with or without external knowledge.

## 6 Conclusion

In this paper, we propose **[RE]VER**🗣: a unified model for **V**erbalizing **E**ntities and **R**elations. We combine definition modeling, relation modeling, and hyper-relation modeling in a unified form and pre-train [RE]VER on a large training data by forming the "entity(s) → sentence" reconstruction task. Extensive experiments on three tasks and six datasets demonstrate the superiority of our model, especially in low-resource settings.

There are various applications of [RE]VER. First, [RE]VER itself can be used as a tool for humans to explore entities and relations by providing interpretable text descriptions, which can help humans better understand entities and relations. This is particularly useful in the scientific domain, where researchers come across new terms every day and want to understand previously unknown concepts and relationships between relevant concepts (Zhu et al., 2023), and in the e-commerce domain, where users want to understand the function of specific products and the relationship between the recommended product and the product he/she already bought (e.g., *tripod* and *camera*) (Huang et al., 2023a). Second, as shown in our experiments, [RE]VER can be applied to improve the performance on entity and relation verbalization tasks such as definition modeling, relation modeling, and generative commonsense reasoning. Third, [RE]VER can serve as a knowledge source to provide knowledge on entities and relations to enhance models designed for entity & relation-related tasks (Ren et al., 2016; Bach and Badaskar, 2007; Lin et al., 2015).

## Limitations

There are two main limitations of this work. First, we do not address ambiguity explicitly in the model training process. During the data preprocessing process, entities are represented by their unique identifiers in Wikidata, eliminating ambiguity and ensuring consistency. However, the input to the models does not include these identifiers (i.e., only the surface name is used). We have chosen this design to increase the system's flexibility, as users are not required to provide identifiers, and the model can handle unseen entities (e.g., those without an identifier). The model may be adapted to deal with ambiguity by including identifiers as part of the input during training.

Second, although continually pretraining enables [RE]VER to generate definitions or relation definitions for entities in a zero-shot setting, the performance still leaves much to be desired. How to further improve the zero-shot performance is an interesting and important future research direction.

## Acknowledgements

We thank the reviewers for their constructive feedback. This material is based upon work supported by the National Science Foundation IIS 16-19302 and IIS 16-33755, Zhejiang University ZJU Research 083650, IBM-Illinois Center for Cognitive Computing Systems Research (C3SR) and IBM-Illinois Discovery Accelerator Institute (IIDAI), grants from eBay and Microsoft Azure, UIUC OVCR CCIL Planning Grant 434S34, UIUC CSBS Small Grant 434C8U, and UIUC New Frontiers Initiative. Any opinions, findings, conclusions, or recommendations expressed in this publication are those of the author(s) and do not necessarily reflect the views of the funding agencies.

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
