# OpenReview forum: "VER: Unifying Verbalizing Entities and Relations"
_EMNLP/2023/Conference — EMNLP 2023 Findings_

### Official Review · Reviewer_hSHA · 2023-08-04

**Soundness:** 4

**Excitement:**

4: Strong: This paper deepens the understanding of some phenomenon or lowers the barriers to an existing research direction.

**Paper Topic And Main Contributions:**

This paper presents a model for verbalising entity and relations, namely generating descriptions of entities and entity relations using pre-trained models.

**Reasons To Accept:**

- (+) Unified approach to different tasks (definition, relation and hyper-relation modeling).
- (+) Thorough evaluation and state-of-the art results.


**Reasons To Reject:**

- (-) None, really. As a comment, I think the paper would improve by including even a small overview of examples of good and bad generations in the appendix.

**Reproducibility:**

4: Could mostly reproduce the results, but there may be some variation because of sample variance or minor variations in their interpretation of the protocol or method.

**Reviewer Confidence:**

5: Positive that my evaluation is correct. I read the paper very carefully and I am very familiar with related work.

---

> ### Author Rebuttal · Authors · 2023-08-27
>
> Thank you for your thoughtful review and positive feedback on our paper. We are glad that the reviewer recognized the benefits of our unified approach for different tasks and valued the comprehensive evaluation.
>
> Thank you for your suggestion to include examples in the appendix. In the current version, we displayed some generation examples from CommonGen in Table 5. We will add more examples in the final version.

---

### Official Review · Reviewer_W9C6 · 2023-08-04

**Soundness:** 4

**Excitement:**

3: Ambivalent: It has merits (e.g., it reports state-of-the-art results, the idea is nice), but there are key weaknesses (e.g., it describes incremental work), and it can significantly benefit from another round of revision. However, I won't object to accepting it if my co-reviewers champion it.

**Paper Topic And Main Contributions:**

This work proposes a unifying model to generate definitions for a set of entities by pre-training encoder-decoder transformers (BART) on a proposed dataset WiD from Wikipedia that has input-output pairs of entity_set-sentences. WiD uses the first sentences of Wikipedia pages for entity definitions and entity-annotated sentences to represent relations between a set of annotated entities. Considering three settings (no fine-tuning, low-resource fine-tuning and full fine-tuning) for three tasks (definition modelling, pairwise relation modelling and hyper-relation modelling for relations between more than two entities), the proposed model generates better definitions than the baselines.

**Questions For The Authors:**

- How did you pre-trained the baseline BART on low-resource settings without using the proposed WiV dataset?
- In line 280, the representation of the input text is unclear. If you encode the input using the set of entities and the augmented sentences, then what is the output text produced by the decoder?

**Reasons To Accept:**

- Well-written and easy-to-follow paper.
- Extensive experiments using three tasks (definition modelling, pairwise relation modelling and hyper-relation modelling for relations between more than two entities) and different settings (no fine-tuning, low-resource fine-tuning and full fine-tuning) and multiple evaluation metrics including human evaluations.
- The obtained quantitative and qualitative results are impressive despite the simplicity of the proposed idea.

**Reasons To Reject:**

- The novelty of the work is limited.
- The importance of unifying the two tasks of defining entities and defining relations between entities is not well-motivated.
- On the challenging setting with more than two entities (hyper-relation modelling), the proposed model shows lower performance than the state-of-the-art model. This indicates the need of using a more informative and structured corpus for relationship definitions between concepts than the proposed dataset.

**Reproducibility:**

3: Could reproduce the results with some difficulty. The settings of parameters are underspecified or subjectively determined; the training/evaluation data are not widely available.

**Reviewer Confidence:**

4: Quite sure. I tried to check the important points carefully. It's unlikely, though conceivable, that I missed something that should affect my ratings.

**Typos Grammar Style And Presentation Improvements:**

- Clarify the difference between bolded and underlined results in the caption of Table 4.

---

> ### Author Rebuttal · Authors · 2023-08-27
>
> We sincerely appreciate your comprehensive review of our paper and your acknowledgment of its clarity, extensive experiments, and impressive results.
>
> **Response to W1 “novelty”**
>
> The beauty and novelty of our approach lies in the idea of unifying and its simplicity paired with its effectiveness, as substantiated by the impressive results highlighted by the reviewer.
>
> **Response to W2 “motivation for unifying tasks”**
>
> First, training and deploying a single model for multiple tasks conserves resources.
>
> Second, as evidenced by the results of VER- and VER in Table 1, integrating the two tasks can also offer slight enhancements to definition modeling.
>
> Third, a unified approach is crucial for applications that require seamless integration of entity definitions and entity relationships. A unified approach ensures consistent representation and understanding of entities and their relationships.
>
> We will emphasize the motivation more explicitly in the final version.
>
> **Response to W3 “performance on hyper-relation modeling”**
>
> Thank you for your comments! We agree that a more informative and structured corpus could enhance performance. However, even without this, our model still achieves performance comparable to SOTA on CommonGen. This is noteworthy, especially considering that the SOTA is specifically tailored for the CommonGen benchmark.
>
> **Response to Q1 “Pre-training of Baseline BART on Low-resource Settings”**
>
> We didn't pre-train the BART baseline from scratch in the low-resource setting. Instead, as indicated in Line 368-371, BART (with the official pretrained weights) was fine-tuned on the low-resource portion of each task-specific dataset, benefiting its pre-existing knowledge gained from large-scale pre-training. We will make this point more clear in the final version.
>
> **Response to Q2 “Representation of Input Text (Line 280)”**
>
> Our encoder takes both the set of entities and the augmented sentences as input. Subsequently, the decoder generates the target sentence containing all entities from the set. As clarified in Line 274, to prevent the target sentence from mirroring the augmented sentence, we exclude the target sentence during retrieval.
>
> **Clarification on Table 4**
>
> The bolded results indicate the best performing methods across the row, while underlined results signify the second-best performance. We will provide a clear description in the caption of the final version.

---

### Official Review · Reviewer_ZaBz · 2023-08-06

**Soundness:** 3

**Excitement:**

3: Ambivalent: It has merits (e.g., it reports state-of-the-art results, the idea is nice), but there are key weaknesses (e.g., it describes incremental work), and it can significantly benefit from another round of revision. However, I won't object to accepting it if my co-reviewers champion it.

**Paper Topic And Main Contributions:**

This paper presents a system that can generate sentences that represent entities and their relationships. It proposes a unified model for verbalizing entities and relations, i.e., generating high-quality sentences describing entities and their relationships. A large dataset of entity sets and sentences is used to pre-train the model in a self-supervised text reconstruction task. The authors also conduct extensive experiments to evaluate the effectiveness of the model, such as definition modeling, relation modeling, and generative commonsense reasoning.


**Questions For The Authors:**

How to detect and mitigate model hallucination?


**Reasons To Accept:**

A dataset was ready for use by the NLP community to follow up this research on entity and relation verbalization tasks, such as definition modeling, relation modeling, and generative commonsense reasoning.

**Reasons To Reject:**

A few limitations need to be addressed as follows.

1. Ambiguity: The authors acknowledge that their model does not explicitly address ambiguity in the training process. This could potentially lead to incorrect or misleading verbalizations, especially when dealing with homonyms or polysemous words.

2. Zero-shot performance: While the model performs well in low-resource settings, its zero-shot performance is not as impressive. This could limit the model's applicability in real-world scenarios where training data may not always be available.

3. Lack of diversity in datasets: The model is trained and evaluated on a limited number of datasets, which may not fully represent the diversity and complexity of real-world entities and relations.

4. Dependence on Wikipedia: The model heavily relies on Wikipedia for pre-training data. This could potentially limit the model's ability to generalize to other domains or types of text.


**Reproducibility:**

4: Could mostly reproduce the results, but there may be some variation because of sample variance or minor variations in their interpretation of the protocol or method.

**Reviewer Confidence:**

2: Willing to defend my evaluation, but it is fairly likely that I missed some details, didn't understand some central points, or can't be sure about the novelty of the work.

---

> ### Author Rebuttal · Authors · 2023-08-27
>
> We greatly appreciate your thorough review and the insights provided. We understand the concerns raised and would like to address them in the following manner:
>
> **Ambiguity:**
>
> As mentioned in our limitation section (Lines 617-625), the model can be adapted to deal with ambiguity by including identifiers as part of the input during training. We opted not to include identifiers to enhance the system's flexibility, as users are not required to provide identifiers, and the model can handle unseen entities (e.g., those without an identifier).
>
> It’s also important to underscore that the expansive dataset our model was trained on equips it to effectively manage a significant majority of potential ambiguities. For instance, in relation modeling for (Apple, Google), the context provided by the word "Google" indicates the intended meaning of "Apple." For definition modeling, users can refine the input to circumvent ambiguity, such as specifying "Apple Inc." or "Apple (company)."
>
> **Zero-shot Performance:**
>
> Our model's performance in low-resource settings serves as evidence of its potential. For zero-shot scenarios, while there's room for improvement, the model still demonstrates considerable proficiency. We have observed that the model, due to its prior extensive training, has a foundational capability that aids in many zero-shot scenarios. As an illustration, for definition modeling, REVER’s zero-shot performance surpasses that of BART trained on the full training data.
>
> **Diversity in Datasets:**
>
> We conducted evaluations across a wide spectrum, spanning six datasets and various configurations including zero-shot, few-shot, and full-data settings. This comprehensive assessment should aptly capture the diversity and intricacy inherent to real-world entities and relationships.
>
> **Dependence on Wikipedia:**
>
> Although we primarily used Wikipedia, it was selected for its comprehensive representation of general knowledge. As showcased in our experiments on CommonGen (Section 4.4), the model isn't restricted to Wikipedia-specific domains. For instance, even though the style of sentences used to pre-train [RE]VER differ from those in CommonGen – consider the example, "A dog leaps to catch a thrown frisbee" – the continual pre-training step still benefits generative commonsense reasoning.
>
> **Regarding Model Hallucination:**
>
> This concern is at the forefront of many generative models, including the most advanced large language models like ChatGPT. While we haven't explicitly detailed a mechanism in our current paper, our REVER training approach significantly curtails hallucination. For instance, the model is less prone to hallucination when working with higher quality retrieved sentences. We firmly believe that enhancing the quality of retrieved sentences is pivotal in reducing hallucinations. We plan to explore this more thoroughly in our future research.

---

### Meta-Review · Area_Chair_VGab · 2023-09-18

**Recommendation:** 4

**Metareview:**

This paper proposed a unified method to address a number of tasks involving entities and relations. The main contribution is the elegance of the model, paired with very positive results in some scenarios. However, the generalizability of the method is not fully proved as demonstrated by mixed results in some scenarios.

---

### Decision · Program_Chairs · 2023-10-07

**Decision:**

Accept-Findings

**Comment:**

This paper proposed a unified method to address a number of tasks involving entities and relations. The main contribution is the elegance of the model, paired with very positive results in some scenarios. However, the generalizability of the method is not fully proved as demonstrated by mixed results in some scenarios.